# Towards Mid-Infrared Gas-Discharge Fiber Lasers

**Alexey Gladyshev** [1], **Dmitry Komissarov** [1], **Sergey Nefedov** [2], **Alexey Kosolapov** [1], **Vladimir Velmiskin** [1], **Alexander Mineev** [2] **and Igor Bufetov** [1,*]

[1] Prokhorov General Physics Institute of the Russian Academy of Sciences, Dianov Fiber Optics Research Center, 38 Vavilov St., 119333 Moscow, Russia; alexglad@fo.gpi.ru (A.G.); komdg@fo.gpi.ru (D.K.); kaf@fo.gpi.ru (A.K.); vvv@fo.gpi.ru (V.V.)

[2] Prokhorov General Physics Institute of the Russian academy of sciences, 119991 Moscow, Russia; nnssmm@yandex.ru (S.N.); mineev@kapella.gpi.ru (A.M.)

[*] Correspondence: iabuf@fo.gpi.ru

**Abstract:** A 2.03 μm gas-discharge fiber laser based on atomic xenon is investigated. Various gas mixtures, such as He–Xe, Ar–Xe, He–Ar–Xe, and He–Ne, are studied by optical emission spectroscopy. The possibility of extending laser generation further into the mid-infrared range is analyzed.

**Keywords:** hollow-core optical fiber; fiber laser; gas-discharge laser; microwave discharge; mid-infrared

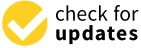



## 1. Introduction

Over the past decade, fiber lasers have made significant progress in accessing the mid-infrared (mid-IR) spectral range [1]. Various approaches have been developed including rare-earth-doped fluoride [2] and chalcogenide [3,4] fiber lasers, chalcogenide-fiber supercontinuum sources [5,6], and laser sources based on hollow-core silica fibers (HCFs) [7–9]. The latter approach is of particular interest since the silica HCFs provide unsurpassed resistance to optical damage, thus opening a promising way towards high power mid-IR fiber lasers.

The introduction of negative-curvature hollow-core silica fibers [10,11], especially those with a simplified design [10,12,13], made low-loss HCFs widely available and led to rapid progress in HCF-based gas fiber lasers (GFLs) emitting at various wavelengths in ~1–5 μm spectral range.

Two types of the GFLs were demonstrated depending on the optical properties of a gaseous active medium that fills the hollow core of a fiber.

The first type of GFLs relies on stimulated Raman scattering (SRS) to convert the pump radiation to longer wavelengths. Due to high values of Raman gain and large vibrational Stokes shifts of the lightest molecular gases ($H_2$, $D_2$, $CH_4$), the radiation of well-developed near-IR lasers can be converted with high quantum efficiency into the mid-IR range by using very few SRS cascades [14–16]. High-power capabilities were also confirmed by demonstrating Raman GFLs operating with average output power as high as 110 [17], 6 [18], and 1.4 W [15] at the wavelengths of 1.15, 2.9, and 4.42 μm, respectively. Moreover, the Raman GFLs enabled a convenient way of generating ultrashort pulses in the mid-IR [14,16,19] and allowed SRS-assisted mid-IR supercontinuum generation in HCFs [20].

The second type of GFLs is based on population inversion, which is achieved between ro-vibrational energy levels of some molecular gas that fills the hollow core. A narrow-linewidth near-IR laser is usually used to pump the gas molecules. As a result, the GFL generates mid-IR radiation. Since 2018, when population inversion GFLs had reached watt-level continuous-wave operation in the mid-IR [21], this type of lasers has shown rapid progress. By using such gases as $C_2H_2$, $CO_2$, and HBr, an output power as high as 8 [22], 3.1 [23], and 0.5 W [24] has been demonstrated in population inversion GFLs at the

wavelengths of 3.1, 4.16, and ~4.3 μm, respectively. The operating wavelength of such GFLs has been extended up to 4.82 μm in CO-filled HCFs [25].

Until recently, optical pumping was used in all gas fiber lasers, based on both SRS and population inversion. In a typical GFL, the pump laser radiation is coupled into a gas-filled HCF in a single-pass configuration (Figure 1a). As a result, laser radiation with longer wavelength appears at the output of the HCF, which essentially serves as a wavelength *converter* only. Importantly, the modern GFL performance, including high-output power capabilities, is limited in this scheme by the solid-state pump lasers, rather than by the hollow-core fibers.

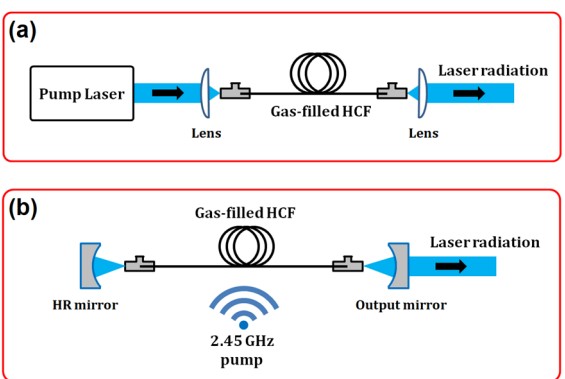

**Figure 1.** Schematic diagrams of (**a**) optically pumped gas fiber laser and (**b**) gas-discharge fiber laser (with microwave pump indicated as an example).

In order to fully realize the capabilities of the hollow-core fibers, the laser radiation should be *generated* directly inside the HCF without the need for a pump laser (Figure 1b). In principle, this task could potentially be accomplished by using gas discharge to pump the GFLs. If developed, the gas-discharge fiber lasers (GDFL) could merge the advantages of both gas lasers and fiber lasers in a single device.

Attempts to develop GDFLs face the problem of ignition and maintenance of a gas discharge in a thin hollow core that is usually as small as ~10–100 μm in diameter. This problem has been investigated in a number of works [26–31]. As a result, a stable gas discharge was successfully demonstrated in glass capillaries [26] and HCFs [27–31] with a hollow core diameter of the order of ~100 μm. To maintain a stable discharge inside gas-filled HCFs, various power supplying schemes were used, including very high DC voltage sources [26,28,29] as well as radiofrequency [30] and microwave (MW) sources [27,30,31]. Optical amplification at several wavelengths in the range of 3.1–3.5 μm was also reported in a He–Xe discharge, which was maintained in an HCF by longitudinally applied 40 kV DC voltage [28,29].

Progress in the generation of laser radiation inside an HCF has been achieved very recently by demonstrating the first GDFLs [32,33]. An MW slot antenna configuration enabled an electric field at a frequency of 2.45 GHz to be transversely applied to the HCF filled by He–Ar–Xe [32] or He–Xe [33] gas mixtures. As a result, not only was stable gas discharge observed, but laser generation was also achieved at the wavelength of 2.03 μm when cavity mirrors were placed at the ends of the HCF. These results open up new avenues for generating laser radiation at various wavelengths from ultraviolet to mid-infrared that are hardly accessible by other methods.

In this work, we investigate the 2.03 μm GDFL based on atomic xenon and analyze the possibility to extend the GDFL operation further into the mid-IR, in particular to 3.51 μm. Optical emission spectroscopy is applied to get insight into the processes of the laser levels population in a gas discharge that is confined in a revolver-type HCF.

## 2. Experimental Setup

In our experiments, we used a revolver-type hollow-core fiber with a hollow core diameter of 130 μm (see Figure 2 on the right). The fiber cladding was formed by 9 capillaries that had a diameter of 40 μm and wall thickness of 2.7 μm. Optical losses calculated for the fundamental mode of the fiber were as low as ~0.01 and ~0.02 dB/m for the wavelengths of 2.03 and 3.51 μm, respectively. Measured by a cut-back method, the optical losses showed higher values (~0.3 dB/m for both 2.03 and 3.51 μm), which was a consequence of multi-mode excitation of the HCF during the measurements. The middle part of the fiber, about 30 cm long, was placed into a slot in the side wall of a metal MW waveguide (see Figure 1). The total length of the revolver HCF was 120 cm. The HCF transmission bands included the most probable wavelengths for obtaining lasing from transitions of neutral xenon atoms, namely, 2.03 and 3.51 μm. At the ends of the revolver fiber, there were miniature vacuum chambers, which made it possible to pre-evacuate the hollow core and then fill it with a selected active mixture of gases at a certain pressure. The design of the vacuum chambers also included adjustable mirrors that formed an optical cavity at wavelengths of 2.03 μm and in a wide range of about 3.5 μm. A more detailed description of the setup can be found in [31,32].

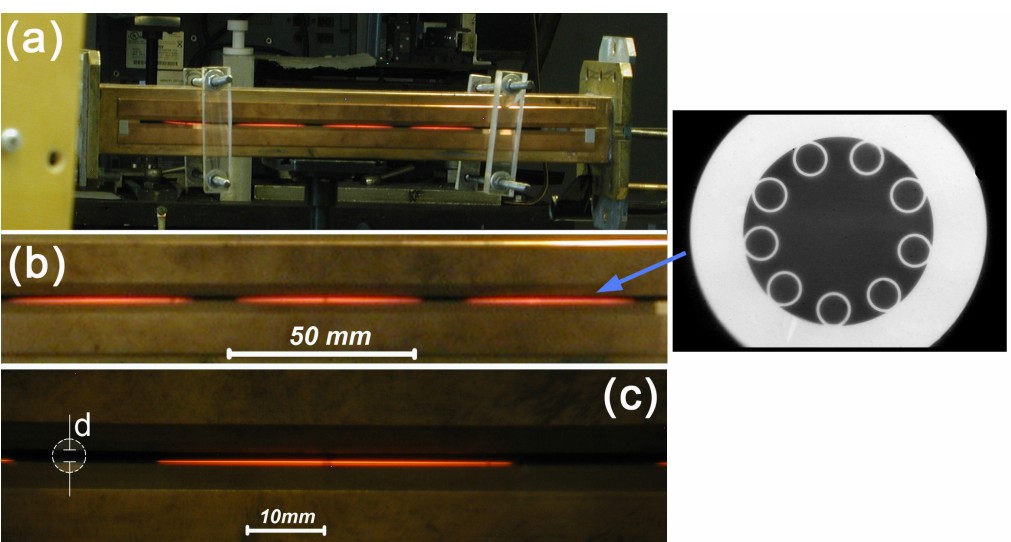

**Figure 2.** (**a**) General view of the experimental setup. The MW waveguide of 90 × 45 mm cross section with a slot in the side wall is shown. The MW discharge can be seen in a slot, where a revolver HCF with a core diameter of 130 μm filled with neon at a pressure of 30 Torr is placed. The periodic structure of the glow is associated with the interference of MW waves propagating along the waveguide. (**b**) Closer view on the glow of MW discharge that is maintained inside the HCF placed into the slot. An image of the HCF cross section obtained with a scanning electron microscope is shown on the right. (**c**) Zoomed view on a single period of the MW discharge glow. The HCF is fixed on the lower edge of the slot, which has a width of d = 2 mm. The outer diameter of the HCF is 300 μm.

To maintain the discharge, MW radiation from a magnetron operating at a frequency of 2.45 GHz in a pulsed mode was used. The pulse repetition rate was chosen to be 400 Hz, and the pulse duration was varied from 20 to 80 μs. The peak power of MW pulses at the input to the MW waveguide was varied from 1.5 to 3.2 kW (the average power did not exceed 100 W). In this case, only a small part of the magnetron power was absorbed in the HCF, since the diameter of the plasma formation was smaller than the diameter of the hollow core (130 μm), and much smaller than the width of the slot (~2 mm). Therefore, most of the MW power was emitted through the slot or returned back to the generator, where it was absorbed in a special protective device. To exploit the MW power in a more efficient way, the microwave waveguide was short-circuited at one end by a metal plunger.

As a result, a standing microwave pattern was formed in the slot, thus allowing roughly doubling the magnitude of electric field for the same MW power. In principle, however, a traveling microwave scheme could also be used in future works.

To facilitate the ignition of the discharge in the HCF, the gas in the fiber core was pre-ionized by the UV radiation of a mercury lamp, which ensured ignition of the discharge when the magnetron was turned on. UV irradiation was turned on only for a few seconds, after which the discharge continued to burn (that is, it was repeatedly self-ignited by each subsequent pump pulse) only under the influence of MW radiation, despite the fact that the time interval between pulses was quite large (2.5 ms). Laser radiation was recorded behind the output mirror of the resonator by using a broadband (1–5 μm) detector based on a photoresistor. The time resolution of the photodetector was ~3 μs. The approximate position of the lasing wavelength was preliminarily determined by using 500 nm bandpass filters, and then more accurately measured by a spectrum analyzer [32].

We also studied the spectral parameters of the MW discharge in the hollow core of the revolver fiber. The spectra of radiation emitted by the MW discharge plasma in the direction perpendicular to the axis of the fiber were recorded. The spectrometer (Ocean SR2, Ocean Optics, Dunedin, FL, USA) made it possible to record spectra in the range from 200 to 1000 nm with a spectral resolution of 0.5 nm.

This setup based on a slot antenna is convenient for experiments in the laboratory. When using GDFLs in practice, of course, other designs will be more suitable, for example, based on an MW strip line [34].

## 3. Results and Discussion

### 3.1. Start of GDFL Generation

The first question we studied was how the gas-discharge fiber laser starts to generate immediately after the pump pulses are applied to the gas-filled HCF. A gas mixture of He:Xe = 100:1 at a pressure of 120 torr was used for the laser emission at 2.03 μm to occur in optimal conditions for quasi-continuous operation [33]. The GDFL pulse peak power was measured as a function of time elapsed from the turning on of the magnetron (Figure 3a). Peak power measurements were taken with a time step of 5 s. It was found that immediately after discharge ignition, the peak power of laser pulses is only ~50% of its maximum steady-state value, which is reached only after several tens of minutes. The observed behavior was typical for the "cold" ignition, i.e., the case where the discharge was not in operation for at least 12 h before the experiment (Figure 3a, curve 1). However, in the case of "warm" ignition, e.g., if only 30 min have passed after the previous experiment, the laser output power is higher at the beginning and reaches the steady-state value somewhat faster compared with the "cold" ignition experiment (Figure 3a, curve 2).

To get more insight into the "cold" ignition process, the temporal shape of the laser pulses was monitored during the experiment (Figure 3b). Although at the very beginning of the discharge, up to 15 successive MW pump pulses were required for the laser generation to appear, this process takes no more than 40 ms, and then, every pump pulse gives rise to the generation of a laser pulse. An important observation was that the optical power was decaying during the pulse. The negative slope of the pulse shape is quite steep at the beginning of the GDFL operation but slowly decreases from pulse to pulse, until the laser operation reaches a steady state, in which the flat-top pulses with maximum power are generated.

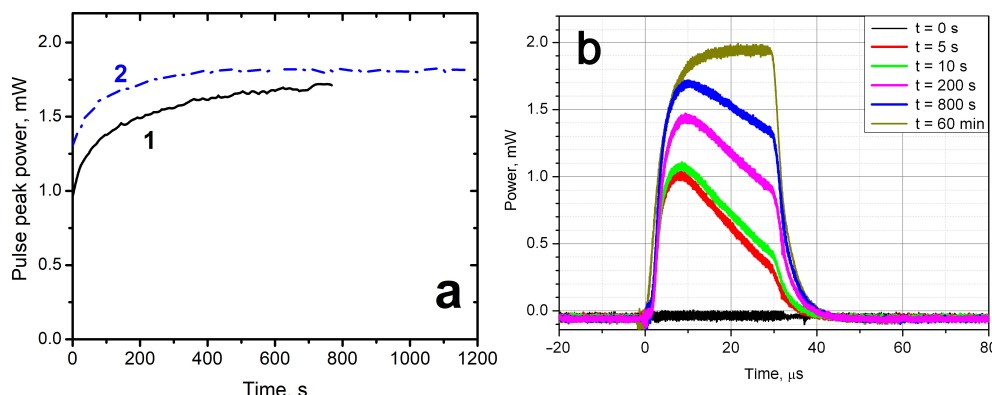

**Figure 3.** (**a**) GDFL pulse peak power vs. time elapsed from the very beginning of discharge ignition (time resolution 5 s). 1—"cold" ignition, 2—"warm" ignition of the MW discharge in the HCF of GDFL. (**b**) Oscillograms of GDFL generation pulses. The gas mixture is He:Xe = 100:1 at a pressure of 120 torr as measured in the HCF and buffer volumes before the magnetron was turned on. The graph indicates the time points to which each curve refers. The duration of the MW pump pulses was 30 μs.

The observed phenomenon could be related to the following factors: (1) the change in gas pressure in the hollow core during the discharge ignition and the subsequent transient process; (2) the change in HCF optical losses that takes place when the gas pressures inside the hollow core and inside the cladding capillaries become different [35,36]; (3) the change in gas composition in the hollow core during the discharge that leads to self-cleaning of the gas mixture from the molecular gas impurities [37]. Let us consider all these factors one by one.

1. The negative slope of the laser pulses could indicate the excessive pressure inside the hollow core, as follows from our previous work [33]. In that work, the He–Xe GDFL operating in steady-state conditions was studied, and the shape of the laser pulses was investigated as a function of gas pressure. Pulse shapes similar to those in Figure 3b were observed only when the gas pressure was above ~150 torr, and the slope was increasing with pressure up to self-termination of laser oscillations at pressures as high as 300 torr [33].

In view of that experiment, the "cold" start of laser oscillations in GDFLs can be understood as follows. Just after the discharge ignition, the pump power deposited to the plasma heats up the gas. The gas pressure immediately increases in that part of the hollow core where the discharge takes place, thus bringing the GDFL out of optimal conditions (120 torr for He:Xe = 100:1 mixture, [33]). Then, the excessive pressure relaxes closer to optimal conditions, since a hollow core of a small volume (~$10^{-3}$ cm$^3$) is connected at both ends to the gas cells of much larger volume (~10 cm$^3$) and other buffer volumes (vessels). As a result, the output laser power can be significantly reduced at the very beginning of GDFL operation-and then the power grows with time until the pressure is stabilized, and a steady state is reached. However, based on the results of [38], the period of time required for the pressure to relax in our experimental conditions is estimated to be as short as ~1 s or less. Importantly, this estimate assumes the ends of the hollow core are not blocked by the cavity mirrors, i.e., the distance between the mirror and the HCF end exceeds the diameter of the hollow core. In the current work, however, the cavity mirrors were butt-coupled to the HCF ends with the aim of reducing optical losses in the cavity. In this case, the mirrors prevent the gas flow out of the hollow core, thus increasing the time required for the gas to reach the equilibrium pressure.

2. Since the too-small diameter of the capillaries prevents them from maintaining the discharge, the discharge ignition takes place in the hollow core only. As a result, the gas pressure difference occurs between the core and capillaries. This pressure difference could lead to significant changes in the HCF optical losses [35,36], thus

influencing the level of output laser power during the transient process that follows the discharge ignition. The relaxation times in this case will be, apparently, of the same order as discussed in the previous paragraph.

3. As it has been recently observed [37], the self-cleaning effect, which consists of removing the molecular gas impurities out of the gas mixture, takes place in the hollow core for our experimental conditions. At the very beginning of the gas discharge, spontaneous emission of such molecular impurities as nitrogen, OH-groups, oxygen, and even hydrogen can be observed from the side surface of the HCF. The emission of these impurities reduces slowly on a time scale of the order of 100 s. In principle, the removal of the impurities could be accompanied by improving the conditions for laser generation, and consequently, the output laser power could show growth during some hundreds of seconds.

All three factors discussed above could have impacts on the observed "cold start" laser behavior either simultaneously or separately. In any case, neglecting these effects could lead, for example, to incorrect measurements of optical amplification in gas-discharge HCF systems.

### 3.2. Possibility of Extending the GDFL Operation to Other Wavelengths

The main question of interest in this work concerns the possibility of extending the GDFL operation to other wavelengths in addition to the previously observed 2.03 μm.

Indeed, atomic Xe has a number of strong 5d→6p transitions emitting in the spectral range from 1.73 to 3.9 μm. Some of these wavelengths are filtered out from the GDFL cavity either by high-loss spectral bands of the HCF, or by spectral selectivity of the output mirror. As a result, the GDFL cavity we use supports oscillations on two strong emission lines of the Xe atom, namely, at the wavelengths of 2.03 and 3.51 μm. In practice, however, the 3.51 μm generation has never occurred in the GDFL in our experiments, although the He–Xe mixture was varied in terms of both mole fraction of xenon (0.15–10%) and total gas pressure (75–300 torr) [33]. With a three-component mixture He:Ar:Xe = 100:10:1 used in the first GDFL demonstration [32], the same outcome was observed, i.e., the laser was generated at 2.03 μm without any signature of the 3.51 μm emission. This case is even more intriguing, keeping in mind that similar gas mixtures in bulk xenon lasers enable optical gain at the wavelength of 3.51 μm to be as high as 50 dB/m [39], which exceeds the gain at 2.03 μm.

To shed light on such a difference in behavior of 2.03 μm and 3.51 μm transitions in the GDFL, the processes of the laser levels population should be examined. Two main possibilities can be considered: (1) fast nonradiative decay of population of the upper laser levels ($5d[3/2]_1^0$ and $5d[7/2]_3^0$), which can be induced by collisions of the excited Xe atoms with other atoms and with the walls of the hollow core, or (2) excessive accumulation of population of the lower laser levels ($6p[3/2]_1$ and $6p[5/2]_2$). Both of these possibilities would reduce the population inversion, thus decreasing the optical gain and suppressing the laser generation. In our experiments, we concentrated on monitoring the relaxation from lower laser levels of the considered laser transitions in Xe I atoms. For this purpose, optical emission spectroscopy is applied as a suitable technique that can provide a rich set of data on plasma properties.

For the 3.51 μm transition, the lower laser level $6p[5/2]_2$ can radiatively decay to a lower energy level $6s[3/2]_2$, thus emitting a line at 904.5 nm (Figure 4). Similar processes for the 2.03 μm laser transition give rise to luminescence at 916.3 nm. As a result, the intensities of luminescent lines at 904.5 and 916.3 nm indicate the populations of lower laser levels for the 3.51 and 2.03 μm transitions, respectively (Figure 4).

Conveniently, both luminescence lines can be detected from a side surface of the HCF and measured with reasonable resolution by a sensitive spectrometer based on the silicon matrix. In this way, optical emission spectra of MW gas-discharge plasma in the hollow core were measured for various gas mixture compositions and pressures. The emission spectra were monitored in the 870–930 nm spectral range, which contains a group of closely

spaced emission lines that is one of the spectral fingerprints of the excited Xe atoms. As a reference, this group of lines was measured for pure Xe MW plasma, showing that in this case, both wavelengths of interest (904.5 and 916.3 nm) have equal intensity in our experimental conditions (Figure 5a, arrows 1 and 2).

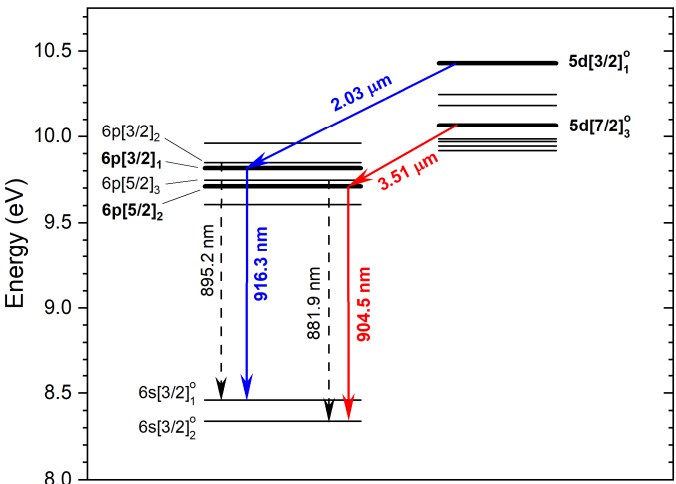

**Figure 4.** Partial diagram of xenon energy levels with the two strongest laser transitions and corresponding transitions from the lower laser levels in the near-IR range.

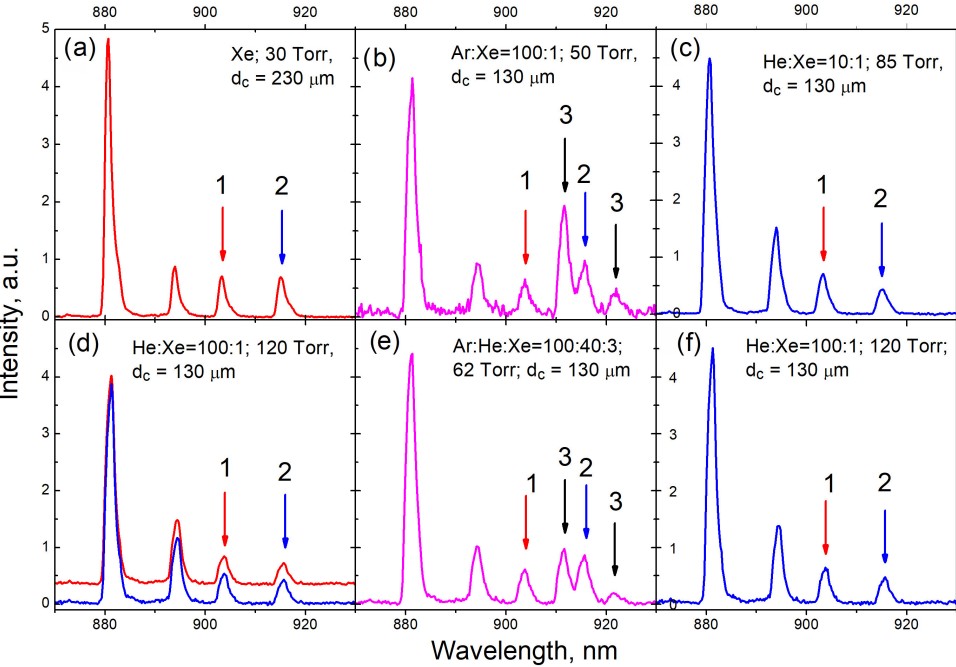

**Figure 5.** Plasma luminescence spectra in the HCF under various experimental conditions when the plasma is maintained by MW radiation. The spectra show a characteristic group of luminescence lines of neutral Xe atoms at wavelengths 881.9 nm; 895.2 nm; 904.5 nm (designated as 1 (red arrow)); 916.3 nm (indicated as 2 (blue arrow)). The emission lines of neutral Ar atoms at 912.3 nm and 922.4 nm are designated as 3 (black arrow). (**a–f**): Plasma spectra in pure Xe and its various mixtures with He and Ar. The compositions of the mixture, the total pressures, and the diameters of the hollow core fiber are indicated on each spectrum. (**d**) shows the spectrum of the discharge in a hollow-core fiber, which is the amplifying element of a He–Xe GDFL, in the absence of laser generation (the cavity is misaligned),the lower curve (blue line), and in the presence of laser generation (upper, red curve).

First, the gas mixture of He:Xe = 100:1 at a pressure of 120 torr was investigated when the GDFL was generating at 2.03 μm in optimal conditions. Importantly, the intensities of all the luminescence lines were not depending on the presence of 2.03 μm generation. When the GDFL cavity was completely misaligned to suppress the laser generation at 2.03 μm, the luminescence spectrum (Figure 5d, lower blue curve) was the same as for the case when the 2.03 μm generation took place in the properly aligned GDFL cavity (Figure 5d, upper red curve). This fact implies that in populating of the lower laser level ($6p[3/2]_1$ in Figure 4), some alternative process prevails, rather than the 2.03 μm laser transition. The role of such a process could be played by electron impact excitation from long-lived lower lying $6s[3/2]_1$ and $6s[3/2]_2$ metastable states [40–43]. This excitation channel increases the population of lower laser levels for both 2.03 μm and 3.51 μm transitions, thus reducing the population inversion on these transitions and trying to suppress the laser generation on both of them. In this situation, depopulation of lower laser levels by atomic collisions becomes important to enable the laser generation.

We observed that for those gas mixtures in which the mole fraction of He atoms prevails, the 916.3 nm emission line (Figure 5, blue arrow 2) becomes less intensive (Figure 5c,f) which may indicate a decrease in the population of the level $6p[3/2]_1$. Thus, high concentration of the He atoms in a gas mixture helps to achieve the population inversion and, consequently, the laser generation at the 2.03 μm transition of atomic Xe.

We noted the obtained result correlates well with the data available in the literature on bulk gas-discharge xenon lasers [40–43]. Moreover, the literature survey indicates that argon atoms are a suitable plasma component for selective depopulation of the $6p[5/2]_2$ laser level. Thus, the gas mixtures with large mole fractions of Ar atoms are promising for the GDFL generation at 3.51 μm.

Our experiments with gas mixtures containing large mole fractions of argon do not reveal any indications that the Ar atoms contribute to depopulation of the lower laser level $6p[5/2]_2$ of the 3.51 μm transition (Figure 5b,e). Actually, the intensity of the 904.5 nm emission line is less compared with the 916.3 nm line. This indicates a shift in the depopulation process towards a greater decrease in the $6p[5/2]_2$ level population, thereby promoting generation at 3.51 μm. Nevertheless, the GDFL generation at 3.51 μm has not been achieved in these conditions. This fact implies that experience accumulated in the field of bulk gas lasers could not be applied in a straightforward way to the GDFLs.

Spectral selectivity of the GDFL cavity we used is also favorable for laser oscillations at the wavelength of 3.39 μm, which can be generated in a He–Ne gas-discharge. Therefore, we filled the RF with a He–Ne gas mixture in a typical ratio of 10:1. Although the MW discharge was observed in the HCF at gas pressures from 60 to 300 torr, the laser generation at 3.39 μm was not achieved.

To get some insight into the processes preventing the He–Ne laser generation, optical emission spectroscopy was applied. In this case, the wavelength of interest was 359.3 nm, which corresponds to luminescence from the lower laser level of the 3.39 μm transition of atomic Ne. The intensity of 359.3 nm emission line is known as a good indicator for the intensity of stimulated emission at the wavelength of 3.39 μm [44,45].

Measured from the side surface of the HCF, a typical emission spectrum of the He–Ne discharge showed absence of the emission line at 359.3 nm (Figure 6a,(a1)). For comparison, the luminescence spectrum of a bulk commercial helium–neon laser (model LGN203) is also shown (Figure 6b). The spectrum is measured from the side surface of the working discharge tube, which had an inner diameter of about 2 mm and contained a gas mixture at a pressure of about 2.5 torr with a composition similar to what we used in the GDFL experiments.

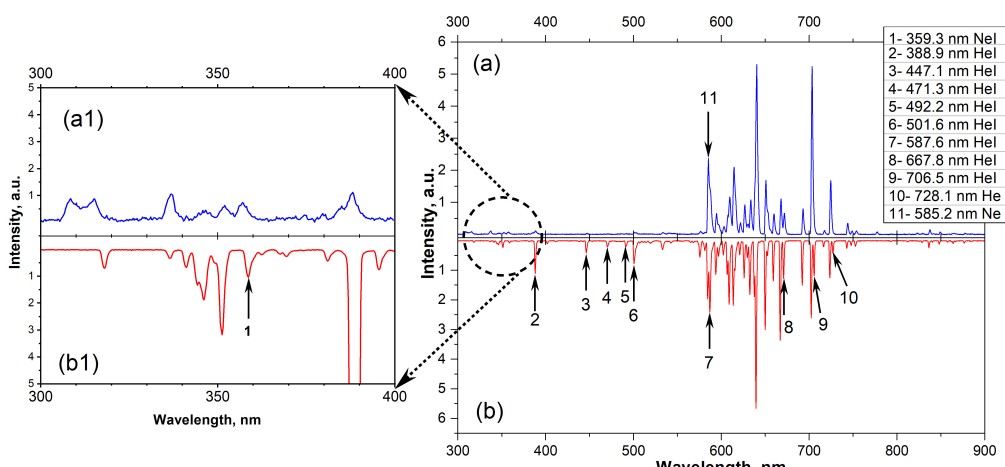

**Figure 6.** (**a**) Spectrum of an MW discharge in an HCF with a core diameter of 130 μm. The core is filled with a He:Ne = 10:1 mixture at a total pressure of 100 Torr, the length of the discharge region is 30 cm. (**b**) Spectrum of a DC discharge in a commercial helium–neon laser LGN203 (discharge tube inner diameter ~2 mm, He:Ne = 10:1 mixture, pressure ~2.5 Torr, discharge length ~30 cm). (**a1**) and (**b1**) are zoomed spectra for the wavelength band of 300–400 nm. The arrows (2–10) in (**b**) indicate He I lines that are absent in spectrum (**a**). The arrow (11) in (**a**) indicates the Ne I line, which is significantly weaker in spectrum (**b**). The arrow (1) in (**b1**) indicates the Ne I luminescence line at a wavelength of 359.3 nm. The inset shows the wavelengths of the spectral lines indicated by arrows.

Note the emission spectrum of the bulk laser tube has a very weak line at 359.3 nm. Although this line is two orders of magnitude lower than the bright He I and Ne I lines (Figure 6b), it indicates the presence of superluminescence at the 3.39 μm laser transition in the bulk cavity. However, in the case of the He–Ne discharge in the HCF cavity, even such a low level of the 359.3 nm emission was not observed.

Comparison of the measured spectra (Figure 6) shows that the luminescence of the He–Ne discharge confined in the RF is presented mostly by the atomic neon lines (Ne I), while emission of the atomic helium lines (He I) is weak. In contrast, the neutral He atoms produce many easily detectable emission lines, when the discharge takes place in the bulk cavity (Figure 6b,(b1), black arrows). Understanding the observed phenomena requires further investigations and could enable the development of compact and efficient gas-discharge fiber lasers in the mid-IR range.

## 4. Conclusions

A new type of lasers, a gas-discharge fiber laser, is studied. It is shown that at the start of the GDFL operation, its optical characteristics drift slowly on a time scale of 10–30 min due to a slow rate of gas-dynamic processes, which is an inherent property of the GDFLs, since the active gas has to flow through a hollow core that is as small as ~100 μm in diameter.

The processes of the laser levels population are studied by optical emission spectroscopy in various gas mixtures, such as He–Xe, Ar–Xe, He–Ar–Xe and He–Ne. It is shown that large mole fractions of the He atoms in the He–Xe and He–Ar–Xe mixtures facilitate depopulation of the lower laser level for the 2.03 μm transition of atomic Xe. As a result, laser generation at the wavelength of 2.03 μm was achieved in these gas mixtures.

In contrast, the gas mixtures with a high Ar content have not provided the laser generation at any wavelength in our experimental conditions. Nevertheless, some indications have been found that collisions with the Ar atoms assist in depopulating of the lower laser level for the 3.51 μm transition of atomic Xe. Thus, Ar-containing gas mixtures remain promising for achieving GDFL generation at the wavelength of 3.51 μm.

The study of the He–Ne gas mixture has revealed very different properties of the gas discharges confined in hollow-core fibers and in the bulk tube of a commercial He–Ne laser.

These findings highlight the need for further research on gas discharges inside hollow-core fibers that have small inner diameters.

The obtained results take a step towards better understanding the properties of gas-discharge plasmas confined in hollow-core fibers. Further research could enable the development of compact and efficient gas-discharge fiber lasers in the mid-IR range.

**Author Contributions:** Conceptualization, I.B. and A.G.; methodology, A.M, S.N., I.B., D.K. and A.G.; formal analysis, I.B.; investigation, S.N., A.G., D.K., I.B. and A.M.; resources, A.K. and V.V; writing—original draft preparation, I.B.; writing—review and editing, A.G., A.M. and I.B.; visualization, A.G.; supervision, A.G., I.B. and A.M.; project administration, A.G.; funding acquisition, A.G. and I.B. All authors have read and agreed to the published version of the manuscript.

**Funding:** This research was supported by Russian Science Foundation (grant № 22-19-00542), https://rscf.ru/project/22-19-00542/.

**Data Availability Statement:** The data supporting the findings of this study are available within the article.

**Conflicts of Interest:** The authors declare no conflicts of interest.

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
