# Peer review of "Towards Mid-Infrared Gas-Discharge Fiber Lasers"

_photonics, doi:10.3390/photonics11030242_

Round 1

Reviewer 1 Report

Comments and Suggestions for Authors

The authors presented the study of the temporal properties of 2.03-µm lasing in a hollow-core fiber-based gas fiber laser and their attempts to further extend such lasing into a longer wavelength regime (3.51 µm). Optical emission spectroscopy was implemented to study the processes of the laser population in various gas mixtures, providing insights into the future development of mid-IR gas-discharge fiber lasers. This work is attractive to the readers of the Photonics journal. I would recommend this work to be published, and I believe this work could be strengthened by addressing the following questions.

1. Fig. 3a: What are the actual peak powers in mW that curves 1 and 2 can reach at their steady state?

2. Fig. 6: I would recommend the authors label each arrow and list their corresponding wavelengths in the caption.

3. What's the transverse mode of the 2.03-µm laser output? I wonder if the authors have measured the beam profile. What is the difference between the cavity loss for 2.03-µm lasing and 3.51-µm lasing? 

Reviewer 2 Report

Comments and Suggestions for Authors

Comments of photonics-2887907

 Mid-IR fiber laser have drawn a lot of interest for its applications in spectroscopy, imaging, and so on. In this manuscript, authors have investigated the 2.03-μm gas-discharge fiber laser based on atomic xenon and analyzed the possibility to extend the GDFL operation further to 3.51 μm.. The results are exciting and this research is worth further investigated. The English writing of this manuscript is very well. I recommend this work for publication in Photonics.

Reviewer 3 Report

Comments and Suggestions for Authors

This paper presents experimental data and discussions on Gas discharge  fiber laser where different gas mixtures were activated by a RF discharge within the hollow core of the fibre. In my view it is a novel and very promising approach for the further development of FL sources.

 I can add just few comments which in my opinion should  be addressed:

1. In the section 2 (experimental setup) more detail should be given on the concrete design of the HCF fibre used (geometry, its capillaries etc) and on the transmission properties of the fibre (optical loss values for the transmission bands- lines 92-95)

2. The fibre excitation pattern in the fig.2, it is consistent with RF standing wave inside the waveguide.It is not clear why authors did not use running wave to get uniform excitation along the fibre. It would be good to include some comments on this.

3. In the part 3.1 authors presented explanation on the "cold start" behavior of their laser by the  excessive gas pressure inside the energized section of the fibre. Here,  I really can not take authors' explanation. The pressure within the fibre should leveled up almost instantaneously (within tens of ms- given really tiny fibre volume). Estimations from Helmholtz cavity model would give resonant frequency of about 100 Hz (though it is usually a bit overestimated). Lines 178-190 of the authors' text are misleading, as we are not considering pressure in large gas volumes (line 184) being equalized through a thin connecting fibre.) All up, the explanation which was presented does not seem to be credible. I would suggest the authors should come up with some alternative theory.

4. In the section 3.2  a detailed discussion on radiation properties of Xe gas and mixtures (and also He-Ne) is  presented . Once it is not really clear why the lasing was not achieved in the 3 micron  wavelength region, authors attempted to explain that by the increase of population of lower laser levels. In my view there could other explanations. For example, broadening of the metastable levels due to collisions of hot ions with fibre walls etc . I suggest the authors should not limit their explanation to one single possibility and extend a bit their discussions on that.
